# Old but New: Group IIA Phospholipase A_2_ as a Modulator of Gut Microbiota

**DOI:** 10.3390/metabo12040352

**Published:** 2022-04-14

**Authors:** Yoshitaka Taketomi, Yoshimi Miki, Makoto Murakami

**Affiliations:** 1Laboratory of Microenvironmental and Metabolic Health Sciences, Center for Disease Biology and Integrative Medicine, Graduate School of Medicine, The University of Tokyo (UTokyo), 7-3-1 Hongo, Bunkyo-ku, Tokyo 113-8655, Japan; taketomiys@m.u-tokyo.ac.jp (Y.T.); miki@m.u-tokyo.ac.jp (Y.M.); 2Lipid Metabolism Project, Center for Basic Technology Research, Tokyo Metropolitan Institute of Medical Science (TMIMS), 1-5-8 Kamikitazawa, Setagaya-ku, Tokyo 156-8506, Japan

**Keywords:** phospholipase A_2_, gut microbiota, metabolomics, lipid metabolism

## Abstract

Among the phospholipase A_2_ (PLA_2_) superfamily, the secreted PLA_2_ (sPLA_2_) family contains 11 mammalian isoforms that exhibit unique tissue or cellular distributions and enzymatic properties. Current studies using sPLA_2_-deficient or -overexpressed mouse strains, along with mass spectrometric lipidomics to determine sPLA_2_-driven lipid pathways, have revealed the diverse pathophysiological roles of sPLA_2_s in various biological events. In general, individual sPLA_2_s exert their specific functions within tissue microenvironments, where they are intrinsically expressed through hydrolysis of extracellular phospholipids. Recent studies have uncovered a new aspect of group IIA sPLA_2_ (sPLA_2_-IIA), a prototypic sPLA_2_ with the oldest research history among the mammalian PLA_2_s, as a modulator of the gut microbiota. In the intestine, Paneth cell-derived sPLA_2_-IIA acts as an antimicrobial protein to shape the gut microbiota, thereby secondarily affecting inflammation, allergy, and cancer in proximal and distal tissues. Knockout of intestinal sPLA_2_-IIA in BALB/c mice leads to alterations in skin cancer, psoriasis, and anaphylaxis, while overexpression of sPLA_2_-IIA in *Pla2g2a*-null C57BL/6 mice induces systemic inflammation and exacerbates arthritis. These phenotypes are associated with notable changes in gut microbiota and fecal metabolites, are variable in different animal facilities, and are abrogated after antibiotic treatment, co-housing, or fecal transfer. These studies open a *new* mechanistic action of this *old* sPLA_2_ and add the sPLA_2_ family to the growing list of endogenous factors capable of affecting the microbe–host interaction and thereby systemic homeostasis and diseases.

## 1. Introduction

The phospholipase A_2_ (PLA_2_) superfamily comprises a group of lipolytic enzymes that typically hydrolyze the *sn*-2 position of glycerophospholipids (hereafter phospholipids) to release free fatty acids (FFAs) and lysophospholipids (LPLs). The mammalian genome encodes more than 50 PLA_2_s or related enzymes, which are classified into several families on the basis of their structures and functions [1]. Historically, PLA_2_ enzymes have long attracted attention as a regulator of the production of lipid mediators (especially arachidonic acid (AA)-derived metabolites called eicosanoids), since they hydrolyze membrane phospholipids to release polyunsaturated fatty acids (PUFAs; including AA) and LPLs, which can act by themselves as bioactive lipids or act as precursors of a wide variety of lipid mediators. It is now obvious that PLA_2_s also participate in membrane homeostasis by regulating phospholipid composition, in energy homeostasis by fueling FFAs for β-oxidation, in microenvironmental regulation of the balance of saturated versus unsaturated FFAs, and so on. Moreover, several enzymes in the PLA_2_ superfamily have phospholipase A_1_, phospholipase B, lysophospholipase, triglyceride lipase, or acyltransferase/transacylase activity in addition to, or rather than, PLA_2_ activity. Thus, one should be careful that the term “PLA_2_” (or the gene name “*PLA2* (in human)*”* or “*Pla2*” (in mouse)) does not always mean that a given PLA_2_ does act as a canonical PLA_2_ to liberate *sn*-2 FFAs and LPLs from phospholipids.

The secreted PLA_2_ (sPLA_2_) family involves structurally related, low-molecular-mass enzymes with a conserved His-Asp catalytic dyad, which ensures a strict Ca^2+^-dependent “PLA_2_” reaction. The mammalian genome encodes 11 sPLA_2_s (catalytically active IB, IIA, IIC, IID, IIE, IIF, III, V, X, and XIIA and inactive XIIB), which are classified structurally into group I/II/V/X, group III, and group XII branches [2]. Individual sPLA_2_s exhibit distinct tissue and cellular expression profiles and exert specific functions in lipid mediator-dependent or possibly lipid mediator-independent fashions. In general, individual sPLA_2_s exert their specific functions within tissue microenvironments where they are locally expressed. Since mammalian cells are relatively resistant to hydrolysis by mammalian sPLA_2_s, it has been proposed that sPLA_2_s act on the plasma membrane of activated, damaged, or dying cells, rather than that of resting cells in a paracrine manner [3,4,5]. In addition, non-cellular phospholipid components, such as dietary food, lipoproteins, lung surfactant, extracellular vesicles (EVs), and membranes of invading microorganisms such as bacteria and possibly fungi and parasites, act as bona fide hydrolytic targets of sPLA_2_s [6,7,8,9,10,11]. In certain situations, sPLA_2_-binding proteins such as PLA2R1 (sPLA_2_ receptor) modulate the functions of sPLA_2_s [12,13]. Various pathophysiological roles of individual sPLA_2_s, as demonstrated by studies employing sPLA_2_ gene-manipulated mice over the past decades, are summarized in our current reviews [14,15,16,17,18].

Historically, sPLA_2_-IB and -IIA are two classical (prototypic) sPLA_2_s originally identified by protein purification in the late 1980s. While sPLA_2_-IB is secreted from the pancreas into the intestinal lumen and acts as a digestive enzyme [19], sPLA_2_-IIA, initially purified from platelets and inflammatory fluids, is the only sPLA_2_ that is abundantly detected in the circulation of patients with inflammation or infection and has been considered to participate in systemic inflammation and antibacterial defense [20]. Recently, we and Boilard’s group have uncovered a novel aspect of sPLA_2_-IIA: this sPLA_2_ expressed in the intestinal Paneth cells contributes to the shaping of the gut microbiota, thereby secondarily affecting systemic events including immunity, allergy, and cancer in proximal and distal tissues. In this article, we summarize a *new* aspect of this *old* sPLA_2_ via the gut microbiota, thus providing an additional insight into the sPLA_2_ research.

## 2. A Long-Lasting Question: Does sPLA_2_-IIA Act as a Regulator of the Gut Microbiota?

The gut microbiota on the epithelial barriers comprises approximately 3 × 10^13^ microbial cells, whose balance can influence the physiological functions of the host [21,22,23,24]. Dysbiosis caused by various environmental factors, such as a high consumption of sugar, fat, or salt, a low consumption of dietary fiber, abuse of alcohol, or medication or antibiotics [25,26], as well as genetic factors such as mutations in host genes that are involved in the epithelial barrier or innate/adaptive immunity [27,28], leads to an increased prevalence of colitis, allergy, metabolic diseases, neurodegeneration, and cancer.

sPLA_2_-IIA is a prototypic sPLA_2_ that is highly induced in various human tissues during inflammation such as rheumatoid arthritis, sepsis, and COVID-19 infection [29,30,31]. It has been well documented that, in addition to its role as an “inflammatory sPLA_2_” that promotes sterile inflammation by mobilizing lipid mediators [32], sPLA_2_-IIA degrades bacterial membranes as a “bactericidal sPLA_2_”, thereby playing a protective role against bacterial infection [33] (Figure 1A,B). Because sPLA_2_-IIA shows high substrate selectivity toward phosphatidylethanolamine (PE), phosphatidylserine (PS), and phosphatidylglycerol (PG) over phosphatidylcholine (PC), phospholipids comprising the plasma membrane of quiescent mammalian cells, where PC is enriched in the outer leaflet, are relatively resistant to sPLA_2_-IIA-mediated hydrolysis [34,35]. When overexpressed or added exogenously at high concentrations, sPLA_2_-IIA is able to hydrolyze phospholipids in activated, damaged, or apoptotic cells, in which PE and PS are exposed on the outer plasma membrane [36,37]. Recently, it has become obvious that sPLA_2_-IIA, as well as several other sPLA_2_s, efficiently hydrolyzes phospholipids in EVs (microparticles and exosomes), which could explain the sPLA_2_-driven generation of lipid mediators during sterile inflammation or other biological events [10,11,32]. In the context of infection, bacterial membranes, which are rich in PE and PG, are superior hydrolytic targets of this enzyme. Indeed, by degrading bacterial membrane phospholipids, sPLA_2_-IIA efficiently kills Gram-positive bacteria, as well as Gram-negative bacteria in the presence of co-factors such as bacterial permeability-increasing protein (BPI) and lysozyme, at physiological concentrations [20,33,38]. Through its potent bactericidal activity, transgenic mice overexpressing human sPLA_2_-IIA (*PLA2G2A^TGN^*) on a C57BL/6 background are protected from infection with Gram-positive bacteria (e.g., *Bacillus anthracis* and *Staphylococcus aureus*) or Gram–negative bacteria (e.g., *Escherichia coli* and *Helicobacter pylori*) [39,40]. Thus, it has been thought that the primary biological role of sPLA_2_-IIA is to protect against bacterial infection (Figure 1B).

Although sPLA_2_-IIA is expressed in various tissues (inflammatory and epithelial cells in particular) of humans and rats, its expression in mice is limited to the intestine (BALB/c and C3H) or not expressed at all (C57BL/6 and 129/Sv) due to a natural mutation [41,42], making the functional analysis of this enzyme using mouse models difficult. Nevertheless, regardless of animal species, sPLA_2_-IIA is commonly and abundantly expressed in intestinal Paneth cells, which secrete a cocktail of antimicrobial peptides such as defensins, cathelicidins, S100 proteins, RNases, and RegIII [43,44]. Antibiotics treatment decreases sPLA_2_-IIA expression in Paneth cells [45], while the colonization of germ-free C3H mice with conventionally housed mice increases its expression [46], suggesting that intestinal sPLA_2_-IIA expression is induced by gut microbiota. The amount of sPLA_2_-IIA in the small intestine of BALB/c mice is estimated to be ~2000 ng/mg tissue [47], a concentration that is sufficient enough to kill Gram-positive bacteria and even Gram-negative bacteria in the presence of other antimicrobial peptides. Because of these facts, it had been hypothesized that sPLA_2_-IIA secreted from Paneth cells into the intestinal lumen may function as a regulator of the gut microbiota, although solid experimental evidence for this hypothesis had been lacking. Recently, our and Boilard’s groups have independently addressed this long-standing question using gene-manipulated mice (knockout and transgenic) for this sPLA_2_ in a back-to-back way [48,49]. As detailed below, the shaping of the gut microbiota by intestinal sPLA_2_-IIA affects several pathologies in distal tissues, including skin cancer, psoriasis, allergy, and arthritis (Figure 1C).

## 3. Lessons from *Pla2g2a*^−/−^ BALB/c Mice

Since the *Pla2g2a* gene is naturally disrupted in C57BL/6 and 129 strains due to a frameshift mutation as mentioned above [42], it had been difficult to assess the precise in vivo functions of endogenous sPLA_2_-IIA using a standard knockout strategy. Taking advantages of BALB/c mice, which have an intact *Pla2g2a* gene [42], we generated *Pla2g2a*^−/−^ mice on this genetic background by backcrossing the mutated *Pla2g2a* allele in C57BL/6 mice onto a BALB/c background. Unlike the situation in humans and rats, where sPLA_2_-IIA is expressed or induced in many tissues, its expression in BALB/c mice is highly restricted to the intestine [43,50]. Beyond this species difference, the *Pla2g2a*^−/−^ BALB/c strain appears to be a good animal model with which to analyze the physiological function of sPLA_2_-IIA endogenously expressed in the intestine. Using this new knockout mouse model, we have recently obtained compelling evidence that intestinal sPLA_2_-IIA is indeed involved in the modulation of gut microbiota, thereby secondarily affecting the pathology of distal skin. 

We found that, despite the restricted expression of sPLA_2_-IIA in the intestine of BALB/c mice, its gene targeting unexpectedly resulted in notable skin phenotypes [48]. In a model of carcinogen-induced skin cancer, the multiplicity and incidence of skin cancer were markedly reduced, with alterations in the immune responses, in *Pla2g2a*^−/−^ mice relative to *Pla2g2a*^+/+^ mice (note that the two genotypes were housed in separate cages; see below). In a model of psoriasis, the ectopic application of imiquimod elicited more severe ear edema, with a greater expression of several psoriasis markers, in *Pla2g2a*^−/−^ mice than in *Pla2g2a*^+/+^ mice. Expression of sPLA_2_-IIA in the small intestine was decreased after treatment with broad spectrum antibiotics or housing in a germ-free facility, suggesting that its expression is induced by gut microbiota, most likely by bacterial-derived, pathogen-associated molecular patterns (PAMPs; e.g., peptideglycan and lipopolysaccharide) which are known to induce sPLA_2_-IIA expression through the NF-κB pathway [20]. Notably, although the abundance of fecal microbiota at the order and family levels was not different between *Pla2g2a*^+/+^ and *Pla2g2a*^−/−^ mice, hierarchical clustering showed clear separation of the gut microbiota into two groups (*Pla2g2a*^+/+^ cluster and *Pla2g2a*^−/−^ cluster) at the genus level. In fact, several bacteria including Gram-positive (*Ruminococcaceae* and *Lachnospiraceae*) and Gram-negative (*Prevotellaceae* and *Helicobacteraceae*) genera were distinctly present in *Pla2g2a*^+/+^ and *Pla2g2a*^−/−^ mice. Importantly, the phenotypes of skin cancer and psoriasis in *Pla2g2a*^−/−^ mice were absent when both genotypes were co-housed in the same cages (which resulted in mixing of the microbiota through coprophagia) or when these mice were housed in a more stringent pathogen-free facility, in which the expression of *Pla2g2a* in WT mice was reduced and the composition of gut microbiota in the two genotypes became largely even if not solely identical. Among several hit bacteria, as noted above, particular bacterial species belonging to the *Helicobacter* and *Ruminococcaceae* showed better correlation with the skin phenotypes in *Pla2g2a*^−/−^ mice. Although the functional linkage between these specific bacteria and the skin phenotypes in *Pla2g2a*^−/−^ mice needs further elucidation, these results provide insight into the notion that the regulation of gut microbiota by sPLA_2_-IIA is associated with the altered sensitivity to skin cancer and psoriasis.

Transcriptome analysis of the intestine, a proximal tissue where sPLA_2_-IIA is expressed, revealed notable alterations in the expression of genes related to epithelial barrier and immunity in *Pla2g2a*^−^^/−^ mice in comparison with *Pla2g2a*^+/+^ mice. Particularly, a number of genes encoding the variable regions of immunoglobulins were markedly changed in *Pla2g2a*^−/−^ mice compared to *Pla2g2a*^+/+^ mice, probably reflecting the distinct antibody responses to commensal microbiota between the genotypes. The reduced expression of several genes related to the anti-inflammatory PPARγ signaling pathway and the increased expression of several proinflammatory genes could partly explain the exacerbated psoriasis and the increased antitumor immunity in *Pla2g*^−/−^ mice. Overall, the lack of sPLA_2_-IIA allows the intestine to remain in a mildly proinflammatory state, which appears to be consistent with the increased proportion of proinflammatory *Helicobacter* and the decreased proportion of anti-inflammatory *Ruminococcaceae* in *Pla2g2*^−/−^ mice. While *Helicobacter* infection is tightly linked to gastric inflammation and cancer [51,52], there is ample evidence that it is protective against asthma and allergy [53,54]. The increase in *Helicobacter* in *Pla2g2a*^−/−^ mice can explain why mouse strains with a mutated *Pla2g2a* gene are more susceptible to intestinal tumorigenesis than those having an intact *Pla2g2a* gene [42] and also accounts for an inverse correlation between *PLA2G2A* expression and gastric cancer in humans [55]. Moreover, *Helicobacter* infection and gastrointestinal inflammation are associated with psoriasis severity [56]. Thus, the effect of sPLA_2_-IIA on *Helicobacter* might be a key determinant that affects disease susceptibility in proximal (intestine) and distal (skin) tissues. In support of this, *Pla2g2a*^−/−^ mice housed in the *Helicobacter*-free animal facility did not display a psoriasis phenotype.

Metabolome analysis revealed that 19 out of 511 plasma metabolites were significantly altered in *Pla2g2a*^−/−^ mice compared with *Pla2g2a*^+/+^ mice (Figure 2A). These metabolites were classified into several groups such as those related to the urea cycle, reactive oxygen species (ROS), and choline metabolism. The urea cycle is associated with T cell immunity [57], and its dysregulation leads to tumor promotion [58]. Plasma levels of various metabolites in the urea cycle were reduced in *Pla2g2a*^−/−^ mice compared to *Pla2g2a*^+/+^ mice (Figure 2A). Increased production of ROS is often associated with inflammation, metabolic diseases, cancer, and aging [59,60], and the *Pla2g2a*^−/−^ plasma contained reduced levels of several ROS-related metabolites (Figure 2A). Furthermore, plasma levels of choline-related metabolites, whose unusual accumulation often causes cellular transformation [61], were reduced in *Pla2g2a*^−/−^ mice (Figure 2A). In addition, several bacteria-specific metabolites such as trigonelline and ectoine, which can affect inflammation and cancer [62], and dicarboxylic acids such as pimelate, sebacate, and azelate, whose levels are associated with the abundance of *Ruminococcaceae* [63], were reduced in *Pla2g2a*^−/−^ mice relative to *Pla2g2a*^+/+^ mice (Figure 2A). Furthermore, the lipidomic profiling of fecal lipids showed notable reductions in several bacteria-specific lipids with anti-inflammatory potential in *Pla2g2a*^−/−^ mice (Figure 2B,C). These bacteria-derived lipids included a class of hydroxyl-, oxo-, and conjugated forms of linoleic acid (LA) metabolites, such as KetoB (10-oxo-octadecanoic acid), KetoC (10-oxo-trans-11-octadecenoic acid), and CLA1/3 (*cis*-9, *trans*-11- and *trans*-9, *trans*-11-octadecadienoic acids), and that of branched or linear fatty acid esters of hydroxy fatty acids (FAHFAs and OAHFAs). Reportedly, long-chain FAHFAs act on the fatty acid receptors GPR40 and GPR120 to exert anti-inflammatory, -oxidant, and -diabetic functions [64,65], and short-chain FAHFAs with acyl α-hydroxy fatty acids (AAHFAs) show inverse correlation with metabolic disease [66]. Although bacteria species generating these unique lipids are currently unknown, the decrease in these anti-inflammatory bacterial lipids may account, at least in part, for the increased antitumor immunity and exacerbated psoriasis in *Pla2g2a*^−/−^ mice. On the other hand, host-derived lipid mediators, mostly produced by fatty acid oxygenation by lipoxygenases or cytochrome P450s, were unchanged or modestly elevated (rather than decreased) in *Pla2g2a*^−/−^ mice than in *Pla2g2a*^+/+^ mice, probably because of the altered inflammatory state in the gut. Collectively, sPLA_2_-IIA deficiency alters the circulating and fecal levels of various hydrophilic and hydrophobic metabolites that could affect cancer and immunity. Because of the alterations of multiple bacteria and metabolites, it is presently difficult to conclusively define the specific bacteria species and metabolite(s) that would be truly responsible for the skin phenotypes in *Pla2g2a*^−/−^ mice. It is likely that the combined actions of these multiple bacteria and metabolites on host immunity and metabolism may underlie the skin cancer and psoriasis phenotypes by the absence of sPLA_2_-IIA.

Allergic responses are known to be profoundly affected by the gut microbiota [67,68,69]. Therefore, we examined the impact of *Pla2g2a* deficiency on passive cutaneous anaphylaxis (PCA), an immediate-type allergic reaction that depends on mast cell degranulation. To this end, *Pla2g2a*^+/+^ and *Pla2g2a*^−/^^−^ mice co-housed (+) or not co-housed (−) in two different animal facilities (TMIMS and UTokyo) were sensitized intradermally with anti-dinitrophenyl (DNP) IgE and then challenged intravenously with DNP-conjugated albumin as an antigen (Ag) together with Evans blue dye. Strikingly, under the co-housing (−) conditions, IgE/Ag-induced PCA reaction was lower at the TMIMS facility (Figure 3A,B), while it was conversely greater at the UTokyo facility, in *Pla2g2a*^−/^^−^ mice than in *Pla2g2a*^+/+^ mice (Figure 3C,D). In agreement, IgE/Ag-treated and even IgE/Ag-untreated *Pla2g2a*^−/^^−^ mice had more degranulated mast cells than the replicate *Pla2g2a*^+/+^ mice in the latter facility, although the total mast cell number was unaffected (Figure 3E,F). In both facilities, however, co-housed *Pla2g2a*^+/+^ and *Pla2g2a*^−/−^ mice exhibited a comparable PCA response (Figure 3G,H). Thus, the allergic response in *Pla2g2a*^−/^^−^ mice is greatly influenced by housing conditions, implying again the involvement of the gut microbiota. Metagenome analysis of the stool revealed that various bacteria genera were commonly increased (Figure 4A) or decreased (Figure 4B) in *Pla2g2a*^−/^^−^ mice relative to *Pla2g2a*^+/+^ mice in both TMIMS and UTokyo facilities. Notably, there were also several bacteria genera whose abundance showed reciprocal patterns (i.e., increased or decreased in one facility and vice versa in the other facility) (Figure 4C,D), suggesting that one or more of these bacteria might be responsible for the contrasting PCA phenotypes between the two facilities (Figure 3). In particular, *Lachnospiraceae FCS020 group* and *UCG-008,* as well as *Ruminococcaceae UCG-013,* were more abundant in the TMIMS facility than in the UTokyo facility; moreover, in the TMIMS facility, *Lachnospiraceae FCS020 group* and *Ruminococcaceae UCG-013* were decreased, while *Lachnospiraceae UCG-008* was increased, in *Pla2g2a*^−/^^−^ mice relative to *Pla2g2a*^+/+^ mice (Figure 4E). Reportedly, the abundance of *Fimicutes* (including *Ruminococcaceae* and *Lachnospiraceae*) can be positively or negatively associated with the severity of allergic diseases [70,71,72,73]. In contrast, *Mucispirillum*, a bacteria genus positively correlated with Th2-related factors in asthma [74], was uniquely abundant only in *Pla2g2*^−/^^−^ mice at the UTokyo facility (Figure 4E), which might be related to the elevated PCA response (Figure 3C,D). Although further studies are needed to clarify whether the changes in these specific bacteria would be indeed responsible for the variable PCA phenotypes in *Pla2g2a*^−/^^−^ mice, these results nonetheless lend additional support to the view that sPLA_2_-IIA shapes the gut microbiota, thereby secondarily affecting mast cell fate and the associated allergic reaction in distal skin and probably in other tissues.

## 4. Lessons from *PLA2G2A^TGN^* C57BL/6 Mice

Along with the study using *Pla2g2a*^−/^^−^ BALB/c mice as described above, Boilard and colleagues performed a complementary study using *PLA2G2A^TGN^* mice in which the human *PLA2G2GA* gene was overexpressed in *Pla2g2a*-null (naturally mutated) C57BL/6 mice [49]. Generally speaking, the results using transgenic mice should be interpreted with caution, as the expression of a given transgene at a super-physiological level, even in tissues or cells where the enzyme is not intrinsically expressed, could result in an artificial phenotype. An advantage of the *PLA2G2A^TGN^* strain is that the transgene contains a part of the promoter region of human *PLA2G2A* gene, which could partially allow the expression profile of sPLA_2_-IIA similar, even if not identical, to that in humans. Although the levels of sPLA_2_-IIA detected in the blood [250 to 2300 ng/mL (up to 8700 ng/mL)] of *PLA2G2A^TGN^* mice are much greater than those measured in healthy individuals [76], they are comparable to those present in patients with severe bacterial infection or sepsis [77,78]. Hence, the use of *PLA2G2A^TGN^* mice has permitted an insight into pathophysiological manifestations that might be relevant to human diseases.

Boilard and colleagues found that aged *PLA2G2A^TGN^* mice spontaneously developed systemic inflammation, with enlargement of the spleen and lymph nodes, increased counts of T and B cells, plasmablasts and granulocytes, and elevated levels of serum IgG and IgA [49]. These events were similarly observed in *PLA2G2A^TGN^* mice lacking *Pla2g4a* or *Alox12*, ruling out the involvement of cytosolic PLA_2_ (cPLA_2_α) and/or 12-lipoxygenase that potentially act downstream of sPLA_2_-IIA in the context of lipid mediator signaling [28]. However, these phenotypes became milder when *PLA2G2A^TGN^* mice were housed in a different facility with a more stringent pathogen-free standard, suggesting the involvement of the gut microbiota. Indeed, a whole-genome shotgun sequencing of the fecal flora distinguished the *PLA2G2A^TGN^* and WT microbiota as a result of sPLA_2_-IIA overexpression. In particular, *Odoribacter, Prevotell*, and *Helicobacter* were more abundant in *PLA2G2A^TGN^* mice, while *Eubacterium*, *Lachnoclostridium*, and *Clostridium* were more abundant in *Pla2g2a*-mutated WT mice. An overall trend is that Gram-negative genera were more abundant in *PLA2G2A^TGN^* mice and Gram-positive genera were more abundant in WT mice, consistent with the view that sPLA_2_-IIA preferentially kills Gram-positive bacteria. Since the increases in *Helicobacter* and *Prevotella* are often associated with inflammatory processes [79,80,81,82], these changes in the gut microbiota may explain, at least in part, the increased systemic inflammation in *PLA2G2A^TGN^* mice.

The gut microbiota can also influence the severity of rheumatoid arthritis [83]. sPLA_2_-IIA is highly induced in rheumatoid arthritis [29,84], and *PLA2G2A^TGN^* mice have higher susceptibility to the K/B × N serum-induced arthritis model [29,85,86]. Beyond the lipid mediator-dependent, proinflammatory action of sPLA_2_-IIA within the arthritic joint (Figure 1A), Boilard and colleagues found that the changes in gut microbiota significantly affected arthritis sensitivity in *PLA2G2A^TGN^* mice. While the depletion of the gut microflora using broad spectrum antibiotics failed to affect the development of arthritis in WT mice, it abrogated the increased sensitivity of *PLA2G2A^TGN^* mice to the arthritis. The concentration of sPLA_2_-IIA in the serum of *PLA2G2A^TGN^* mice was markedly increased in arthritic mice, whereas the microbiota depletion cancelled this event. Furthermore, while the fecal transfer from *PLA2G2A^TGN^* mice did not affect the severity of arthritis in WT mice, that from WT mice into *PLA2G2A^TGN^* mice reduced the severity, suggesting that the exacerbation of arthritis in *PLA2G2A^TGN^* mice depends largely on the changes in the gut microbiota and that some metabolites present in the WT microbiota could be protective against arthritis. Metagenome analysis of fecal microbiota under the arthritis conditions showed that *Muribaculum* might be associated with the arthritic phenotype of *PLA2G2A^TGN^* mice.

Although there was a limited impact of sPLA_2_-IIA on the gut eicosanoid profile of the host, untargeted lipidomics revealed that sPLA_2_-IIA expression led to significant alterations in the fecal lipid profile. Among the lipid species detected thus far, cholesteryl ester, ether-linked diacylglycerol, and triacylglycerol, as well as multiple LPLs and phospholipids, were elevated in both arthritic and non-arthritic *PLA2G2A^TGN^* mice. The increase in the concentration of total FFAs and LPLs was evident in both arthritic and non-arthritic *PLA2G2A^TGN^* mice, which was mitigated upon the depletion of the microbiota by antibiotics. Using machine learning, lipid metabolites belonging to the diacylglycerol, triacylglycerol, FFA, and ceramide families were found to be associated with sPLA_2_-IIA expression. Overall, these data suggest that sPLA_2_-IIA offers systemic effects on the immune system through its activity on the microbiota and thereby lipid metabolites.

## 5. Summary and Future Prospects

In this article, we have made an overview of the *new* role of an *old* sPLA_2_, sPLA_2_-IIA, as a regulator of gut microbiota, as revealed by its gene targeting and transgenic overexpression recently performed by our and Boilard’s groups, respectively, in a back-to-back methodology [44]. Since sPLA_2_-IIA has the ability to degrade bacterial membranes at physiological concentrations [20,33,38], and since it is abundantly expressed in intestinal Paneth cells that secrete various antimicrobial peptides [43,44], it had long been speculated that this bactericidal sPLA_2_ might play a role in controlling gut microbiota, although there had been no solid experimental evidence using proper animal models. It is now evident that several phenotypes in *Pla2g2a*^−^^/^^−^ and *PLA2G2A^TGN^* mice are profoundly influenced by forcible manipulations (e.g., co-housing, fecal transfer, and antibiotics) of the gut microbiota. The results obtained from the studies by us and Boilard’s group are not necessarily complementary, since they were carried out using different disease models in knockout versus transgenic mice on different genetic backgrounds (BALB/c versus C57BL/6, respectively) using different approaches under different housing conditions in different facilities [44]. Nonetheless, both studies have reached the same conclusion that intestinal sPLA_2_-IIA acts as a host factor that contributes to the shaping of the gut microbiota, whose perturbation by knockout or overexpression leads to systemic effects. To further strengthen this concept, it would be important to identify which bacterial species and/or metabolites altered by sPLA_2_-IIA deficiency or overexpression are responsible for the disease phenotypes in the future study.

Intestinal expression of sPLA_2_-IIA is markedly reduced by treatment with antibiotics or by housing in a germ-free facility, indicating a feed-forward cycle of the sPLA_2_-IIA–microbiota interaction, where some bacteria-derived PAMPs induce the expression of sPLA_2_-IIA, which then shapes the microbial community and thus affects disease severity in distal tissues. It should be noted that the impact of sPLA_2_-IIA on gut microbiota cannot be simply explained by its ability to directly kill bacteria only, since only limited bacteria genera were changed, while the abundance of most bacteria were barely affected by its deficiency or overexpression. It is thus likely that the alteration of gut microbiota in *Pla2g2a*^−/^^−^ or *PLA2G2A^TGN^* mice may also rely on complex interactions between different bacterial species and between bacteria and host. It is conceivable that sPLA_2_-IIA may increase certain bacteria through eradicating competing commensals. Indeed, in the lung of cystic fibrosis patients, Gram-negative *Pseudomonas aeruginosa* (sPLA_2_-IIA-resistant) increases the expression of sPLA_2_-IIA, which in turn kills Gram-positive *Staphylococcus aureus* (sPLA_2_-IIA-sensitive), allowing the former bacterium to be dominant [87]. Alternatively, indirect actions of sPLA_2_-IIA on the microbiota via modulation of the host immunity involving the generation of certain lipid mediators, or via the PLA2R1-driven mechanism, should also be taken into account, even though the overall changes in host lipid mediators in the stool of *Pla2g2a*^−/^^−^ or *PLA2G2A^TGN^* mice were only modest [48,49]. Using *PLA2G2A^TGN^* mice, Schewe et al. reported that sPLA_2_-IIA modifies the YAP-dependent differentiation of intestinal stem cells through the pathway involving the interaction of sPLA_2_-IIA with PLA2R1 [44].

Since several sPLA_2_ isoforms are expressed in the gastrointestinal tract [16,88,89], they may play a general role as a regulator of gut microbiome. Indeed, knockout of pancreatic sPLA_2_-IB, a digestive sPLA_2_ that degrades dietary and biliary phospholipids in the intestinal lumen [19], impairs the defense against parasite infection in association with the change in gut microbiota [90], although it remains unclear whether this event would be related to the improved atherosclerotic and metabolic phenotypes in *Pla2g1b*^−/^^−^ mice [6,91,92]. sPLA_2_-V secreted from macrophages increases phagocytosis and promotes bacterial or fungal clearance [93,94]. Therefore, sPLA_2_-V might also contribute to the clearance of some bacteria or fungi in the gut. Since the deficiency of sPLA_2_-IID, an immunosuppressive sPLA_2_ that is expressed in dendritic cells and M2 macrophages, increases adaptive immune responses against coronavirus infection [95], this sPLA_2_ might also be associated with the intestinal antimicrobial immunity. sPLA_2_-X is another sPLA_2_ that is abundantly expressed in the intestine, especially in colonic epithelial cells and goblet cells [88,96]. Although *Pla2g10*^−/^^−^ mice independently generated and analyzed by different laboratories have been reported to display inflammatory, cardiovascular, and metabolic phenotypes, the results are not necessarily consistent among the studies; for instance, *Pla2g10*^−/^^−^ mice have greater [88] and lower [97] inflammation, increased [98] and decreased [96] body weight, and exacerbated [99] and ameliorated [100] atherosclerosis. Hence, we speculate that the variable phenotypes in *Pla2g10*^−/^^−^ mice observed in current studies might also involve the action of sPLA_2_-X on the gut microbiota. Furthermore, this action mode might be applicable not only to the microbiota in the gut, but also to those in other anatomical sites such as the skin. For instance, sPLA_2_-IIE and -IIF, which are highly expressed in the hair follicle and epidermis, respectively [101,102], might affect skin homeostasis partly by modulating the skin microbiota.

Taken together, the sPLA_2_/microbiota axis described here has opened a new insight into the mechanistic actions of the sPLA_2_ family. We speculate that some of the functions of sPLA_2_s that have been reported to date might be mediated, at least in part, through their actions on the gut microbiota in either a direct (i.e., sPLA_2_ kills bacteria) or indirect (i.e., sPLA_2_ regulates host immunity that in turn affects the microbiota) way. Translation of these findings using mouse models to human pathology is not so simple, since, besides the species difference in sPLA_2_ expression profiles, we need to consider the tissue-intrinsic effects of sPLA_2_s and the extrinsic effects of the gut sPLA_2_/microbiota axis, both of which can be diversely affected by environmental factors. Nonetheless, the findings reported herein have raised the possibility that the increased levels of sPLA_2_-IIA or possibly other sPLA_2_s in human stool could have a predictive value for several diseases. Moreover, oral application of sPLA_2_ inhibitors could be a potential therapy to treat or prevent allergy, arthritis, and cancer. In this regard, oral application of methyl indoxam, a pan-sPLA_2_ inhibitor, efficiently suppressed diet-induced obesity and glucose intolerance in mice [103]. In the context of this review, it is possible that this beneficial effect of the sPLA_2_ inhibitor on systemic metabolism may rely partly on the suppression of the sPLA_2_–microbiota interaction in the gastrointestinal tract, although other mechanisms, such as the inhibition of dietary phospholipid digestion by digestive sPLA_2_s [6,7] or lipoprotein modification by metabolic sPLA_2_s [8], should also be considered.

## Figures and Tables

**Figure 1 metabolites-12-00352-f001:**
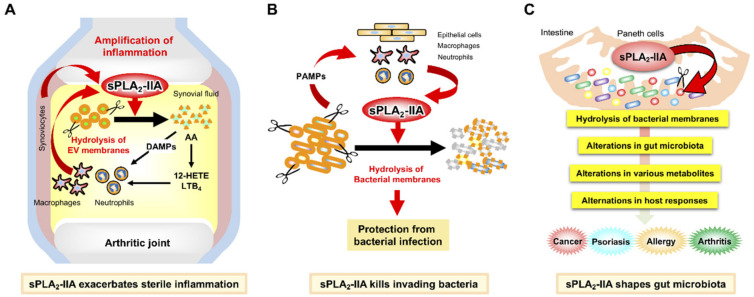
Functions of sPLA_2_-IIA in sterile inflammation, anti-bacterial defense, and microbiota regulation: (**A**) In human rheumatoid arthritis, sPLA_2_-IIA induced in leukocytes and synoviocytes by inflammatory cytokines acts on phospholipids in EVs to release AA, which is further metabolized to eicosanoids such as 12-hydroxyeicosaenoic acid (HEPE) and leukotriene B_4_ (LTB_4_) to promote neutrophil recruitment and activation. sPLA_2_-IIA also releases mitochondrial DNA, one of the danger-associated molecular patterns (DAMPs), from the EVs. As such, sPLA_2_-IIA amplifies sterile inflammation as an “inflammatory sPLA_2_” [10,32]. (**B**) During bacterial infection, sPLA_2_-IIA induced in various cells (such as epithelial cells and leukocytes) by pathogen-associated molecular patterns (PAMPs) (such as peptidoglycan and lipopolysaccharide) kills invading bacteria by degrading bacterial membranes. As such, sPLA_2_-IIA plays a role in host defense against bacterial infection as a “bactericidal sPLA_2_” [20,33]. (**C**) In the intestinal lumen, sPLA_2_-IIA secreted from Paneth cells acts as an antimicrobial protein to shape the gut microbiota, thereby secondarily affecting host responses including cancer, psoriasis, allergy, and arthritis (see text).

**Figure 2 metabolites-12-00352-f002:**
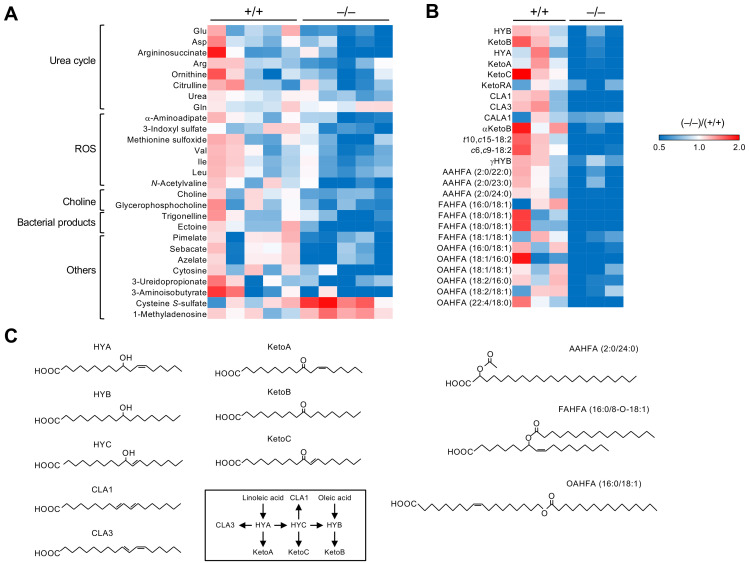
Alterations in the levels of plasma and fecal metabolites in *Pla2g2a*^−/−^ BALB/c mice. The heatmaps indicate fold changes of hydrophilic metabolites in plasma (**A**) and bacterial lipids in stools (**B**) of individual *Pla2g2a*^−/−^ mice compared to *Pla2g2a*^+/+^ mice. (**C**) Structures of representative bacterial lipids listed in (**B**). Biosynthetic routes of hydroxy-, oxo-, and conjugated linoleic acid metabolites are also shown. The mice were housed in a pathogen-free facility (TMIMS) with a 12 h light–dark cycle at 23 °C, with free access to water and food, and all animal experiments were carried out according to protocols approved by the Institutional Animal Care and Use Committees under the Japanese Guide for the Care and Use of Laboratory Animals [48].

**Figure 3 metabolites-12-00352-f003:**
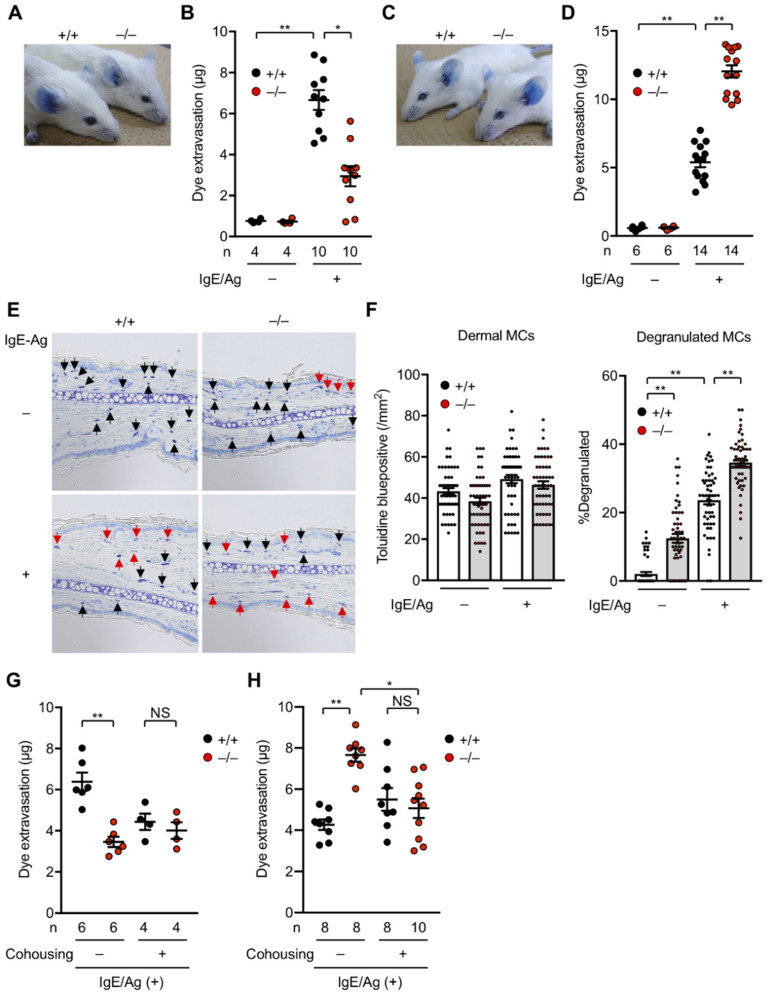
Altered PCA responses in *Pla2g2a*^−^^/^^−^ BALB/c mice under distinct housing conditions. Ears of *Pla2g2a*^+/+^ and *Pla2g2a*^−^^/^^−^ mice (8–12 weeks of age, male) were sensitized by subcutaneous injection of anti-DNP IgE monoclonal antibody (30 ng) and then challenged by intravenous injection of a mixture of DNP-conjugated human serum albumin (60 µg) as an antigen (Ag) together with 1 mg Evans blue, as described previously [75]: (**A**–**D**) Representative photos of the ears (**A**,**C**) and quantification of dye extravasation (**B**,**D**) in IgE/Ag-treated (+) or -untreated (−) *Pla2g2a*^+/+^ and *PLa2g2a*^−^^/^^−^ mice housed in the TMIMS (**A**,**B**) and UTokyo (**C**,**D**) facilities. (**E**,**F**) Histology of the skin (**E**) and quantification of total and degranulated mast cells (**F**) in IgE/Ag-treated or -untreated *Pla2g2a*^+/+^ and *Pla2g2a*^−^^/^^−^ mice housed in the UTokyo facility. The ear pinnae were fixed with 10% (*v*/*v*) formalin, embedded in paraffin, sectioned (4-µm thickness), and stained with toluidine blue. A total of 55 views for each group (*n* = 5). Black and red arrows indicate non-degranulated and degranulated mast cells, respectively. Scale bar, 25 µm. (**G**,**H**) IgE/Ag-induced PCA reaction in *Pla2g2a*^+/+^ and *Pla2g2a*^−/−^ mice with (+) or without (−) co-housing in the TMIMS (**G**) and UTokyo (**H**) facilities. Values are mean ± SEM. *, *p* < 0.05; **, *p* < 0.01; NS, not significant. Statistical analysis was performed using Graph Pad PRISM with Brown–Forsythe test and then Kruskal–Wallis and Dunn’s post hoc test (**B**,**D**,**F**,**G**,**H**). The numbers of mice used for the analysis are indicated in each panel.

**Figure 4 metabolites-12-00352-f004:**
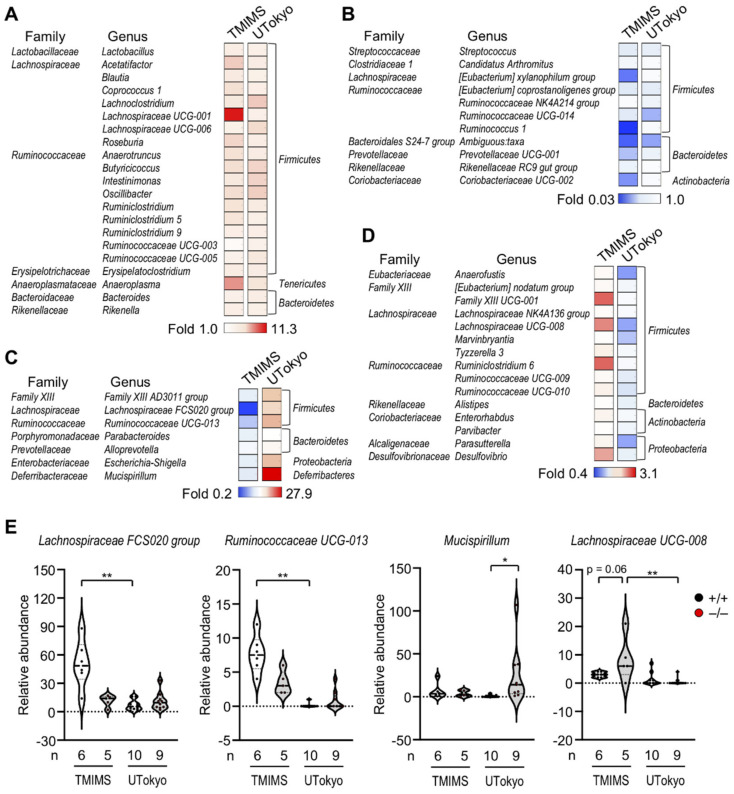
The effects of sPLA_2_-IIA deficiency on the gut microbiota in the TMIMS and UTokyo animal facilities. Metagenome analysis of bacterial 16S RNA in the stool was performed as described previously [48]: (**A**,**B**) Heatmap representation of bacterial genera commonly increased (**A**) or decreased (**B**) in *Pla2g2a*^−/^^−^ mice relative to *Pla2g2a*^+/+^ mice in both facilities. (**C**,**D**) Heatmap representation of bacterial genera that were increased in the UTokyo facility and decreased in the TMIMS facility (**C**) and vice versa (**D**) in *Pla2g2a*^−/^^−^ mice relative to *Pla2g2a*^+/+^ mice. (**E**) Violin plots of the abundance of representative bacteria in *Pla2g2a*^+/+^ and *Pla2g2a*^−^^/^^−^ mice. Values are mean ± SEM. *, *p* < 0.05; **, *p* < 0.01. Statistical analysis was performed with one-way ANOVA followed by Šidák multiple comparison test, or Kruskal–Wallis test followed by Dunn’s multiple comparison test (**E**). The numbers of mice used for the analysis are indicated in each panel.

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
