# Peer review of "Old but New: Group IIA Phospholipase A2 as a Modulator of Gut Microbiota"

_metabolites, 2022, doi:10.3390/metabo12040352_

Round 1

Reviewer 1 Report

Comments on the paper:

Title reflects the paper’s content.

Abstract - requires a  correction (too general)

Introduction

The introduction makes a proper introduction to the subject matter of the paper.

Materials and Methods

The study was carried out using modern and well chosen methods, which guarantee the reliability of the results obtained. Statistical analyzes used in the experiment need to be more detailed. Did the authors assessed if the values obtained had normal distribution before using parametric test? If so, please state how this assessment was made.

Results

Figures and description of results are clear and understandable.

Discussion

Well written.

Conclusion

Conclusions ? (What conclusions have you drawn from it?)

Strong points:

It is worth noting that the laboratory determinations were performed carefully and the analytical procedures were correct. The paper is clear and concise. The study was conducted appropriately and the results are correctly interpreted.

Author Response

Answers to Reviewer 1

[Answer] Thank you very much for your very positive comments. Our point-by-point replies below.

Title reflects the paper’s content.

[Answer] Thank you.

Abstract - requires a correction (too general)

[Answer] We have added the following sentences to the abstract.

“Knockout of intestinal sPLA2-IIA in BALB/c mice leads to alterations in skin cancer, psoriasis and anaphylaxis, while overexpression of sPLA2-IIA in Pla2g2a-null C57BL/6 mice induces systemic inflammation and exacerbates arthritis. These phenotypes are associated with notable changes in gut microbiota and fecal metabolites, are variable in different animal facilities, and are abrogated after antibiotics treatment, co-housing or fecal transfer.”

Introduction

The introduction makes a proper introduction to the subject matter of the paper.

[Answer] Thank you.

Materials and Methods

The study was carried out using modern and well chosen methods, which guarantee the reliability of the results obtained. Statistical analyzes used in the experiment need to be more detailed. Did the authors assessed if the values obtained had normal distribution before using parametric test? If so, please state how this assessment was made.

[Answer] We have added more information on statistical analyses to the legends of Figures 3 and 4. We believe that normal distribution was confirmed by these methods.

Results

Figures and description of results are clear and understandable.

[Answer] Thank you.

Discussion

Well written.

[Answer] Thank you.

Conclusion

Conclusions ? (What conclusions have you drawn from it?)

[Answer] We have modified the 1stparagraph of the “Summary and future prospects” to make the conclusion of this study clearer, as follows.

“In this article, we have made an overview of the newrole of an oldsPLA2, sPLA2-IIA, as a regulator of gut microbiota, as revealed by its gene targeting and transgenic overexpression recently performed by our and Boilard’s groups in a back-to-back way. Since sPLA2-IIA has the ability to degrade bacterial membranes at physiological concentrations and since it is abundantly expressed in intestinal Paneth cells that secrete various antimicrobial peptides, it had long been speculated that this bactericidal sPLA2might play a role in controlling gut microbiota, although there had been no solid experimental evidence using animal models. It is now evident that several phenotypes in Pla2g2a-/-and PLA2G2ATGNmice are profoundly influenced by forcible manipulations (e.g., co-housing, fecal transfer, and antibiotics) of the gut microbiota. The results obtained from the studies by us and Boilard’s group are not necessarily complementary, since they were carried out using different disease models in knockout versustransgenic mice on different genetic backgrounds (BALB/c versusC57BL/6, respectively) using different approaches under different housing conditions in different facilities. Nonetheless, both studies have reached the same conclusion that intestinal sPLA2-IIA acts as a host factor thatcontributes to shaping of the gut microbiota, whose perturbation by knockout or overexpression leads to systemic effects. To further strengthen this concept, it would be important to identify which bacterial species and/or metabolites altered by sPLA2-IIA deficiency or overexpression are responsible for the disease phenotypes in the future study.”

Strong points:

It is worth noting that the laboratory determinations were performed carefully and the analytical procedures were correct. The paper is clear and concise. The study was conducted appropriately and the results are correctly interpreted.

[Answer] Thank you.

Reviewer 2 Report

The manuscript submitted by Yoshitaka Taketomi et al. has summarized recent studies on the roles of group IIA Phospholipase (sPLA2-IIA) in shaping gut microbiota and affecting systemic events including immunity, allergy, and cancer in proximal and distal tissues. There are the following areas to be revised for publishing in our journal.
Major revisions:
1. The authors can consider including more case studies/examples to demonstrate the roles of sPLA2-IIA as a modulator of gut microbiota.
2. The molecular mechanisms underlying the roles of sPLA2-IIA are not sufficiently covered.
3. A summary of current understandings of sPLA2-IIA in killing bacteria and regulating host immunity with more details would be helpful for the readers better understand its role as a modulator of gut microbiota.
4. Page 4 line 160, there is a typo and one of the two Pla2g2a-/- should be Pla2g2a+/+.
5. Characterization of microbiota dynamics in combination with lipidomics could be valuable for mechanistic understandings of the roles of sPLA2-IIA as a modulator of gut microbiota.

Author Response

Answers to Reviewer 2

The manuscript submitted by Yoshitaka Taketomi et al. has summarized recent studies on the roles of group IIA Phospholipase (sPLA2-IIA) in shaping gut microbiota and affecting systemic events including immunity, allergy, and cancer in proximal and distal tissues. There are the following areas to be revised for publishing in our journal.

[Answer] Thank you for reviewing our manuscript. Our point-by-point replies below.

Major revisions:
1. The authors can consider including more case studies/examples to demonstrate the roles of sPLA2-IIA as a modulator of gut microbiota.

[Answer] Unfortunately, there are no more case studies/examples to demonstrate the roles of sPLA2-IIA as a modulator of gut microbiota. We have included all information reported so far (plus a few additional unpublished data to further strengthen the manuscript) in this review.

To answer this reviewer’s comment, we have added historical information of sPLA2-IIA to the 2ndand 3rdparagraphs in the Introduction (highlighted in red), so that general readers would understand why the concept “sPLA2-IIA and microbiota” had emerged.

We believe that summarizing this new role of sPLA2-IIA in this review is timely and contributes to the research field of lipid biology in the context of the thematic issue “Multipurpose Enzymes in Lipid Metabolism”.

2. The molecular mechanisms underlying the roles of sPLA2-IIA are not sufficiently covered.

[Answer] We apologize our insufficient description of the molecular mechanisms underlying the roles of sPLA2-IIA. To address this criticism as much as possible, we have added the following points to the revised manuscript.

Current understandings of sPLA2-IIA in killing bacteria and regulating inflammation have been added to the 2ndand 3rdparagraphs to the Introduction (highlighted in red). We believe that this provides the historical background on the proposal regarding the roles of sPLA2-IIA as a microbiota regulator.

Possible connection of specific bacteria (i.e., Helicobacter) with phenotypes of sPLA2-IIA-deficient mice have been discussed in more detail in the 3rdparagraph of section 3 (highlighted in red), as follows.

“While Helicobacterinfection is tightly linked to gastric inflammation and cancer, there is ample evidence that it is protective against asthma and allergy. The increase of Helicobacterin Pla2g2a-/-mice can explain why mouse strains with a mutated Pla2g2agene are more susceptible to intestinal tumorigenesis than those having an intact Pla2g2agene and also account for an inverse correlation between PLA2G2Aexpression and gastric cancer in humans. Moreover, Helicobacterinfection and gastrointestinal inflammation are associated with psoriasis severity. Thus, the effect of sPLA2-IIA onHelicobactermight be a key determinant that affect disease susceptibility in proximal (intestine) and distal (skin) tissues. In support of this, Pla2g2a-/-mice housed in the Helicobacter-free UTokyo animal facility did not display a psoriasis phenotype.”  

Possible connection of specific metabolites with phenotypes of sPLA2-IIA-deficient mice have been discussed in the 4thparagraph of section 3 (highlighted in red), as follows.

“The urea cycle is associated with T cell immunityand its dysregulation leads to tumor promotion. Plasma levels of various metabolites in the urea cycle were reduced in Pla2g2a-/-mice compared to Pla2g2a+/+mice (Figure 2A). Increased production of ROS is often associated with inflammation, metabolic diseases, cancer, and aging, and the Pla2g2a-/-plasma contained reduced levels of several ROS-related metabolites (Figure 2A). Furthermore, plasma levels of choline-related metabolites, whose unusual accumulation often causes cellular transformation, were reduced in Pla2g2a-/-mice (Figure 2A). In addition, several bacteria-specific metabolites such as trigonelline and ectoine, which can affect inflammation and cancer, and dicarboxylic acids such as pimelate, sebacate, and azelate, whose levels are associated with the abundance of Ruminococcaceae, were reduced in Pla2g2a-/-mice relative to Pla2g2a+/+mice (Figure 2A).”

“Although bacteria species generating these unique lipids are currently unknown, the decrease of these anti-inflammatory bacterial lipids may account, at least in part, for the increased anti-tumor immunity and exacerbation of psoriasis in Pla2g2a-/-mice.On the other hand, host-derived lipid mediators, mostly produced by fatty acid oxygenation by lipoxygenases or cytochrome P450s, were unchanged or modestly elevated (rather than decreased) in Pla2g2a-/-mice than inPla2g2a+/+mice, probably because of the altered inflammatory state in the gut.”

The following point was already written in the last part of the 5thparagraph in section 3.

“Because of alterations in multiple bacteria and metabolites, it is presently difficult to conclusively define the specific bacteria species and metabolite(s) that would be truly responsible for the skin phenotypes in Pla2g2a-/-mice. It is likely that the combined actions of these multiple bacteria and metabolites on host immunity and metabolism may underlie the skin cancer and psoriasis phenotypes by the absence of sPLA2-IIA.”

We have added the following sentences to the last part of the 1stparagraph in section 5 (highlighted in red).

“To further strengthen this concept, it would be important to identify which bacterial species and/or metabolites altered by sPLA2-IIA deficiency or overexpression are responsible for the disease phenotypes in the future study.”  

3. A summary of current understandings of sPLA2-IIA in killing bacteria and regulating host immunity with more details would be helpful for the readers better understand its role as a modulator of gut microbiota.

[Answer] A summary of current understandings of sPLA2-IIA in killing bacteria and regulating host immunity have been added to the 2ndand 3rdparagraphs to the Introduction (highlighted in red).

4. Page 4 line 160, there is a typo and one of the two Pla2g2a-/- should be Pla2g2a+/+.

[Answer] We apologize this careless mistake. We have corrected it.

5. Characterization of microbiota dynamics in combination with lipidomics could be valuable for mechanistic understandings of the roles of sPLA2-IIA as a modulator of gut microbiota.

[Answer] We agree that characterization of microbiota dynamics in combination with lipidomics could be valuable for mechanistic understandings of the roles of sPLA2-IIA as a modulator of gut microbiota. The best way to approach this important issue would be to transfer candidate bacteria or lipids into Pla2g2a-/-or PLA2G2ATGNmice to examine the rescuing effects of the phenotypes. However, because multiple bacteria and metabolites were altered in these mice compared with their WT controls, such an approach will take considerable time and should be performed in the future work. The main purpose of this manuscript is to review current studies, focusing on two papers recently published in the JCI insightin a back-to-back way, thereby appealing the novel action mode of sPLA2-IIA in the special issue of “Multipurpose Enzymes in Lipid Metabolism”.  

Although neither our study (using Pla2g2a-/-mice) nor Boilard’s study (using PLA2G2ATGNmice) reached the identification of key bacteria and metabolites, these two studies are the first to provide strong evidence that sPLA2-IIA acts as a host factor that is primarilyexpressed in the intestine and contributes to shaping of the gut microbiota, whose perturbation by genetic deletion or overexpression culminates in systemic effects. We think that this is an important and novel message. It is possible that the cooperative action of multiple bacteria/metabolites on immunity and metabolism might underlie the systemic phenotypes resulting from sPLA2-IIA deletion or overexpression. We believe that summarizing these two studies is a good opportunity to make a significant advance in the research fields of PLA2/lipid biology as well as microbiome biology.

Currently, sPLA2s have been considered to exert their functions in tissue microenvironments where they are intrinsically expressed. The two back-to-back studies have changed this concept; sPLA2-IIA modulates the pathology of distal tissues by shaping of the gut microbiota. Further, this finding raised the intriguing possibility that some if not all functions of other sPLA2isoforms might also be attributable to their effects on the gut microbiota, a view that is under investigation in our laboratory using KO mice for individual sPLA2s.These points were already described in the 3rdparagraph of section 5.

Reviewer 3 Report

In this review manuscript, the authors summarize a new aspect of the old sPLA2 via the gut microbiota, providing an additional insight into the sPLA2 research, combined their own latest research work and others. Overall, this review is well organized and written, and it should be meaningful and helpful to the related researchers and readers in this field. Here, I only have a small question: “the authors only summarized some work about mice in section 3 (Lessons from Pla2g2a-/- BALB/c mice) and section 3 (3. Lessons from PLA2G2ATGN C57BL/6 mice, I believe it should be section 4), does this mean that there is no research progress in human? If there are any research work or progress in human, please lend a single section to give the related summary. 

Author Response

Answers to Reviewer 3

In this review manuscript, the authors summarize a new aspect of the old sPLA2 via the gut microbiota, providing an additional insight into the sPLA2 research, combined their own latest research work and others. Overall, this review is well organized and written, and it should be meaningful and helpful to the related researchers and readers in this field. Here, I only have a small question: “the authors only summarized some work about mice in section 3 (Lessons from Pla2g2a-/- BALB/c mice) and section 3 (3. Lessons from PLA2G2ATGN C57BL/6 mice, I believe it should be section 4), does this mean that there is no research progress in human? If there are any research work or progress in human, please lend a single section to give the related summary. 

[Answer] Thank you very much for your very positive comments.

First of all, we apologize our careless mistake. We have corrected into section “4.”Lessons from PLA2G2ATGN C57BL/6 mice, andaccordingly, section “5.”summary and future prospects.

We understand that human relevance is very important. Unfortunately, however, there is no research work or progress in human. As already written in the last paragraph in “summary and future prospects”, translation of these findings using mouse models to human pathology is not so simple, since besides the species difference in sPLA2expression profiles, we need to consider the tissue-intrinsic effects of sPLA2s and the extrinsic effects of the gut sPLA2/microbiota axis, both of which can be diversely affected by environmental factors. Nonetheless, the findings reported hereinhave raised the possibility that the increased levels of sPLA2-IIA or possibly other sPLA2s in human stool could have a predictive value for several diseases. Moreover, the oral application of sPLA2inhibitors could be a potential therapy to treat or prevent allergy, arthritis, cancer, and possibly other diseases.

Round 2

Reviewer 2 Report

The authors have addressed all of my previous concerns. However, there are some formatting issues for Figures 1, 2, and 4 which should be solved before publication.

Author Response

Answers to Reviewer 2

The authors have addressed all of my previous concerns. However, there are some formatting issues for Figures 1, 2, and 4 which should be solved before publication.

[Answer] Thank you for reviewing our manuscript. We have reformatted the figures. They look fine on my PC. However, if the formatting of the figures would be still inappropriate, we would like to ask the editorial office to correct them. We think that the reformatting is an editorial issue.